



# A probabilistic seabed-ice keel interaction model

Frédéric Dupont[4], Dany Dumont[1], Jean-François Lemieux[2], Elie Dumas-Lefebvre[1], and Alain Caya[3]

[1]Institut des sciences de la mer de Rimouski, Université du Québec à Rimouski, Rimouski QC, Canada.
[2]Recherche en Prévision Numérique Environnementale/Environnement et Changement Climatique Canada, 2121 route Transcanadienne, Dorval QC, Canada.
[3]Recherche en assimilation de données et météorologie satellitaire, Environnement et Changement Climatique Canada, 2121 route Transcanadienne, Dorval QC, Canada.
[4]Service Météorologique Canadien, Environnement et Changement Climatique Canada, 2121 route Transcanadienne, Dorval QC, Canada.

**Correspondence:** Frédéric Dupont (frederic.dupont@ec.gc.ca)

**Abstract.** In some coastal regions of the Arctic Ocean, as well as in shallow seasonally ice-covered seas, grounded ice ridges contribute to stabilizing and maintaining a landfast ice cover. Recently, a grounding scheme representing this effect on sea ice dynamics was introduced and tested in a coupled ice-ocean model. This grounding scheme, based on a parameterization of ridged keel thickness linearly correlated to the mean thickness, improves the simulation of landfast ice in many regions such as in the East Siberian Sea, the Laptev Sea and along the coast of Alaska. Nevertheless, this parameterization is based solely on the mean properties of sea ice. Here, we extend the parameterization by taking into account subgridscale ice thickness distribution and bathymetry distributions, which are generally non-normal, and by computing the maximum seabed stress as a joint probability interaction between the ice and the seabed. The probabilistic approach shows a reasonably good agreement with observations and with the previously proposed grounding scheme while potentially offering more physical insights in the formation of landfast ice.

## 1 Introduction

Landfast ice is sea ice that remains immobile despite external forces acting on it. It occurs because of two different processes. One is when sea ice is constrained by the land boundaries, i.e. when it is thick and solid enough that it resists compressive, tensile and shear stresses arising from integrated external forces. In narrow channels of the Canadian Arctic Archipelago (CAA), in fjords or in between nearby Islands, landfast ice occurs because the tensile strength of the ice everywhere in between land masses, including at the wall boundaries, is sufficient to resist along-channel wind and current forces. This mechanism has been identified as the main cause for landfast ice in the Kara Sea (Olason, 2016), Nares Strait between Greenland and Canada (Dumont et al., 2009; Plante et al., 2020) or the CAA (Lemieux et al., 2016) using sea ice models having either a plastic or elasto-brittle rheology. Note that this phenomenon can also be simulated using discrete element models with friction and cohesion in between particles (e.g. Garcimartín et al., 2010).



The second distinctive process is in open coastal regions where sea ice is attached to the land on one side, but has a free edge adjacent to the ocean. In order to maintain the stability of landfast ice covers extending over large distances from the coast, horizontal cohesive strength is not sufficient. Sea ice must be grounded, i.e. some of the deformed and thicker ice present in
the ice cover must reach down to the seabed, and thanks to interactions with seabed, provides the required additional resisting force.

Most often than not, the two processes are intertwined, as once an anchor point through grounding is in place, some of the internal horizontal stress is redirected there. The most striking examples of this are the extensive landfast ice that forms off the northern coast of Siberia, in the Laptev and East Siberian Seas (L15), as well as along the Alaskan coast, in the Beaufort Sea
(Mahoney et al., 2007, 2014).

To better simulate landfast ice in the Arctic, Lemieux et al. (2015) (hereafter referred to as L15) —and some adaptations in Lemieux et al. (2016) for the Arakawa B grid staggering)— have developed a parameterization for estimating the seabed stress due to grounded ridges[1]. It estimates the thickness of the largest ridges within a model grid cell as a linear function of the mean thickness, which will be referred in the following as the LKD (linear keel depth) parameterization. When the mean
thickness is greater than a critical value that is a function of the local water depth, the parameterization assumes that ridge keels deep enough to reach the sea floor exist. In such a case, a non-zero basal stress is added to the ice momentum equation. The maximum basal stress that can be supported by the grounded ridge is a function of the weight of the ridge in excess of hydrostatic balance. This parameterization has the practical advantage of being simple and easy to add in single and multi-category models. Moreover, model simulations ran over a pan-Arctic domain with that parameterization show that seabed-ice
interactions lead to realistic landfast ice covers along the Alaskan and Siberian coasts (L15; Lemieux et al., 2016).

However, as recognized by L15, there are a few caveats to that model. The first one is that it assumes that ridges thicker than the mean ice thickness exist, irrespective of the ice formation and deformation history. This is of course not physically adequate as there are situations where the ice thickness mostly results from thermal growth and much less from mechanical deformation. This is particularly true for landfast ice. For example, in a situation where newly formed ice becomes landfast as it consolidates
in relatively shallow waters, the mean thickness will mainly increase due to freezing at the base, or snow conversion to ice at the top. As sea ice remains immobile, no ridges are formed. However in this specific case, the parameterization assumes that ridge keels always exist, which can lead to an overestimation of the seabed-ice interaction. Despite the fact that the LKD parameterization can be tuned with observations of landfast ice in terms of extent and timing, the predictive capability of such a model is thus questionable in a context of rapid climate changes that affect ice thickness, ice types and ice dynamics.
The second caveat of the LKD parameterization is that the keel depth varies linearly with the mean ice thickness, with a tunable proportionality constant approximately equal to 8. However, many observational studies show that the thickness of the deepest keels depend nonlinearly on the mean thickness. For example, Melling and Riedel (1996) report observations of ice draft in the Beaufort Sea that suggest a power-law dependence of the keel draft to the surrounding level ice thickness with an exponent of 0.5. Using submarine sonar observations in Davis Strait, Wadhams et al. (1985) show that the ice thickness

---

[1]The expression used in L15 was *basal* stress, but we find it somewhat ambiguous as other processes happens at the base of the ice that are unrelated to interactions with the seabed





distribution of ice thicker than $3-4$ m follows a negative exponential. More recently, Haas et al. (2010) presented results from a 2400-km long pan-Arctic airborne survey over old ice in the Arctic Ocean, between Svalbard and Alaska, and confirmed that the ice thickness distribution is generally skewed with an exponential tail for values larger then $3-4$ m. Positively skewed ice thickness distributions means that extreme values are not proportional to the mean, but rather non-linearly correlated to it.

Most contemporary sea ice models use an ice thickness distribution (ITD) to parameterize thermal processes and to describe
the subgrid scale thickness variability arising from deformations. Flato and Hibler III (1995) explored in detail how the parameterizations of mechanical redistribution in a 28-category model influence the resulting ITD in various places of the Arctic Ocean. The model confirmed that positive skewness of the ITD is ubiquitous. However, the ITD is not typically resolved with such a large number of categories in climate models or short-term forecasting systems (5-10 categories are usually the norm). Thus the tail of the distribution is still poorly represented in models. Nonetheless, we can approximate analytically the form of
the tail and provide a parameterization that represents more faithfully the thickest keels and their interactions with the seabed.

The main objective of this paper is to layout a probabilistic representation of the seabed-ice interaction that takes advantage of the evolutive ITD, and moreover, of the high-resolution bathymetric information that is typically smoothed to build a model bathymetry. This way, the stress arising from seabed-ice interaction would depend more on bathymetric data and dynamical processes affecting the ITD and would be less dependent on empirical tuning. The structure of the paper is as follows. Sec-
tion 2 presents a derivation of the probabilistic seabed-ice stress, and section 3 describes the numerical implementation and experiments. Results are presented in section 4 and discussed in section 5.

## 2 The probabilistic approach

The sea ice thickness field is highly heterogeneous due to mechanical processes such as compressive ridging that thickens the ice, and crack opening that creates ice-free areas in which new ice can form. Since the pioneering work of Thorndike
et al. (1975), these processes are explicitly represented in sea ice numerical models through the evolution of an ice thickness distribution (ITD) usually called $g(h)$, where $h$ is the ice thickness. As noted by Thorndike et al. (1975), $g(h)$ can be interpreted as a probability density function (PDF) that gives the likelihood that a point will have a thickness $h$. In this sense, the ice thickness is interpreted as a random variable $x$, which has a PDF $g(x)$ that is supposed to be representative of the ice thickness distribution over of certain area — corresponding to the grid cell area in the context of a numerical model. In the rest of the
paper, we will use the symbol $h$ when we refer to statistical moments of the distribution. As we are interested to know if and how sea ice interacts with the seabed, we will assume that the seabed depth in a given grid cell is also represented by a random variable $y$ characterized by the probability density function $b(y)$. Ice of thickness $x$ touches the seabed if the draft of the ice, which is function of ice thickness that we call $D(x)$, is larger than the height of the water column $y$.

Generally, $D(x)$ represents the hydrostatic equilibrium of the floating ice under snow loading. However in the following,
we will discard the snow contribution and consider the simpler relationship $D(x) = \rho_i x/\rho_w$, where $\rho_i$ and $\rho_w$ are the ice and water densities, respectively. This relationship is easily invertible and allows to simplify the equations.





Assuming that both variables are independent, the probability that the ice is in contact with the seabed is noted as $\mathcal{P}(D(x) > y)$ and is obtained by

$$\mathcal{P}(D(x) > y) = \int_0^\infty \int_0^{y=D(x)} g(x)\, b(y) \; dy\, dx \tag{1}$$

If we assume that $g(x)$ and $b(y)$ represent respectively the ITD and the water column height distribution over a surface area $S$, then $S\mathcal{P}(D(x) > y)$ represents the total area of ice that is in contact with the seabed and that can potentially exert friction inside the surface $S$. Now the maximum frictional stress of a flat block of ice of thickness $x$ sitting on the ocean floor at a depth $y$ is equal to $\mu_s F_N(x, y)$, where $\mu_s$ is a static friction coefficient and $F_N(x, y)$ is the normal force exerted by the excess weight of the ice sitting on the ocean floor, normalized over a unit surface area of $1\ \mathrm{m}^2$. Neglecting once again the effect of snow, this force is expressed as

$$F_N(x, y) = \hat{g}\left(\rho_i x - \rho_w y\right), \tag{2}$$

where $\hat{g}$ is the gravitational acceleration. Considering that $x$ and $y$ are fully characterized by their respective probability density functions, the total maximum friction stress is obtained by integrating over all water depth lower than $D(x)$ and over all thickness values, yielding

$$\tau_b^{\mathrm{max}} = \mu_s \hat{g} \int_0^\infty \int_0^{y=D(x)} \left(\rho_i x - \rho_w y\right)\, g(x)\, b(y)\, dy\, dx. \tag{3}$$

This expression differs from the one proposed by L15 (their Eq. 21) in two ways. First, the LKD parameterization prescribes one keel depth for each mean thickness value, while in the probabilistic formulation, the solution is degenerated, which means that there can be multiple keel depth values associated with different ITD that have the same mean thickness value. The second difference is that the keel depth values depend non-linearly on the mean thickness, as opposed to LKD where the prescribed dependence is linear. This will be further illustrated in section 3.1 when discussing the ITD.

To facilitate the numerical implementation, we follow L15 and the instantaneous horizontal seabed-ice stress vector $\boldsymbol{\tau}_b$ is written as a function of the sea ice velocity $\mathbf{u} = u\hat{\mathbf{i}} + v\hat{\mathbf{j}}$ as

$$\boldsymbol{\tau}_b = -\tau_b^{\mathrm{max}} \left(\frac{u\hat{\mathbf{i}} + v\hat{\mathbf{j}}}{|\mathbf{u}| + u_0}\right) e^{-\alpha_b(1-A)}. \tag{4}$$

where $\alpha_b$ is a constant controlling the dependence of the seabed stress on the ice concentration $A$ and $u_0$ is a small velocity scale that assures a smooth transition between static and dynamic regimes. As in L15, it is also assumed that the kinetic friction coefficient is equal to the static one (i.e., $\mu_s$). The seabed stress vector is then included in the momentum equation as

$$m_i \frac{D\mathbf{u}}{Dt} = -\mathbf{k} \times m_i f \mathbf{u} + \boldsymbol{\tau}_a + \boldsymbol{\tau}_w + \boldsymbol{\tau}_b + \nabla \cdot \boldsymbol{\sigma} - m_i \hat{g} \nabla \eta \tag{5}$$

where $\boldsymbol{\tau}_a$ is the wind stress, $\boldsymbol{\tau}_w$ is the water stress, $f$ is the Coriolis parameter, $m_i$ is the combined mass of ice and snow per unit surface, and $\eta$ is the sea surface elevation. In the next section, we describe how Eq. 3 is computed, and we emphasize these differences by comparing the results using LKD as well as with observations.





## 3 Probability density distributions and numerical implementation

Solving for the seabed stress using the probabilistic approach presented in the previous section requires that distributions are known. In principle, these can have arbitrary shapes if the integral of Eq. 3 is computed numerically, which is done here (see section 3). The following sections present how distributions of ice thickness, or ice draft, and bathymetry are determined, how

their parameters are optimized to represent seabed-ice interactions, and how sensitive the resulting seabed stress is to these parameters.

### 3.1 Ice thickness distribution

The number of thickness categories sea ice models typically use is a trade-off between the representativity of ice dynamics and thermodynamics, and computational cost. Sea ice components of Earth climate modeling systems typically use five categories,

which is the default number of the Los Alamos Community Ice (CICE) model (Hunke et al., 2021). The Canadian Regional Ice-Ocean Prediction System (RIOPS, described in Dupont et al. (2015) and Smith et al. (2021)) uses 10 categories. In all cases, thickness categories are chosen to represent the positively skewed distributions. However, their number is generally too low to resolve the exponentially-decreasing tail of the distribution, and thus to make a precise assessment of the deepest keels that are the most likely to interact with the seabed. Figure 1a shows an example of an ITD discretized as in RIOPS with 10 categories

with levels at 0, 0.1, 0.15, 0.3, 0.5, 0.7, 1.2, 2.0, 4.0 and 6.0 m. All categories are populated with ice. The last category includes ice thicker than 6.0 m and is in theory unbounded (note that the upper bound is arbitrarily set at 10 m in the plot). In CICE, the mean thickness inside each category is in practical terms defined as the ratio between the partial ice volume divided by the partial concentration of a particular populated category. This thickness varies between the bounds of the category following thermodynamic processes, advection and mechanical ridging. However, ice properties can be transported in thickness space out

of a given category to adjacent ones regardless of the relative position of this mean category thickness inside the bounds. The actual PDF of ice thickness inside each category is approximated by a delta function, a uniform distribution or a decreasing exponential depending on the user's input, and sometimes on the considered process (as shown below).

In order to represent the tail of the distribution and the depth of the largest keels, we proceed in two steps. The first step consists in computing the mean and variance of the discretized (model) ITD and find a positively skewed PDF — let's call

it $f(x)$ — that has the same mean and variance. This ensures that the PDF conserves the same total concentration and mean ice thickness. The second step consists in identifying the percentile that best represents the thickness of the deepest keel for this PDF, and then truncate the PDF to that value. This is necessary because PDFs are usually defined from 0 to infinity, and because the contribution of small probabilities for extreme values are amplified by the term $(\rho_i x - \rho_w y)$ in Eq. 3, which may result in an overestimation of the seabed stress. Note that the truncation to a certain percentile does not conserve the total

integrated concentration nor the volume. In the following, we will provide some estimation of this error.

There is no general consensus on the functional form that the ITD should follow, and no model reproduces exactly observed ITDs. Wadhams et al. (1985) suggested that ice thicker than 2-4 m can often be fitted by a negative exponential, while Flato and Hibler III (1995) used a power law. Toppaladoddi and Wettlaufer (2015) assumed that ridged ice formation follows a random





process that is similar to the Brownian motion that obeys an advection-diffusion Fokker-Planck equation for which the solution
has a negative exponential. Roberts et al. (2019), using variational principles, macroporosity and a Mohr-Coulomb ridging
rheology, show results that slowly converge to a negative exponential. However, despite the lack of consensus on the functional
form, the positive skewness is generally accepted as a property that a representative ITD should have.

The function we use here is the log-normal distribution that is both representative of observed ITDs with negative exponential
tails, and easily manipulable numerically. It is given by

$$f_{\mathcal{LN}}(x) = \frac{1}{\sqrt{2\pi}\sigma x} \exp\left(\frac{(\ln x - \mu)^2}{2\sigma^2}\right) \tag{6}$$

where $\mu$ and $\sigma$ are determined from the mean $m$ and variance $v$ by the following expressions

$$\mu = \ln\left(\frac{m}{\sqrt{1 + v/m^2}}\right) \tag{7}$$

$$\sigma = \sqrt{\ln(1 + v/m^2)}. \tag{8}$$

The percentile value $x_p$ corresponds to the value below which a percentage $p$ of the thickness values falls and is given by

$$x_p = \exp\left[\mu + \sqrt{2\sigma^2}\, \mathrm{erf}^{-1}(2p - 1)\right]. \tag{9}$$

The value of $x_p$ should represent the thickness of the deepest keel one can find in an ice cover characterized by the correspond-
ing ITD, and is a parameter to which the seabed stress parameterization is very sensitive. How this value is tuned is presented
in section 3. Once $x_p$ is determined, $f_{\mathcal{LN}}(x)$ is truncated so that values $x > x_p$ do not contribute to the integral of Eq. 3. Hence,
the error introduced in truncating the integral using $p =$0.997 is 0.3% in terms of concentration, which is by definition equal
to $1 - p$. The mean thickness of the the truncated ITD in Fig. 1c is equal to 2.13 m, which corresponds to a 2% difference
compared to the full distribution. These errors decrease of course with increasing value of $p$.

Figure 1b shows a second example of an ITD where all categories are populated except the very last one. In this case, the
redistribution scheme of the model did not produce ice thicker than 6 m (the maximum value of the 9[th] category). Consequently,
the log-normal PDF associated to that ITD is truncated at 6 m.

## 3.2 Bathymetry distribution

The bathymetry field of coupled ice-ocean numerical models is typically built from gridded datasets such as GEBCO (Becker
et al., 2009) or ETOPO (Amante and Eakins, 2009), which are themselves aggregations of data acquired from many different
sources. In some places, the density of data points is much higher than the model grid resolution, while in some sparse areas,
data is absent and some level of interpolation is required. The method proposed here would allow for taking into account the
subgrid scale distribution of the bathymetric field. However, instead of creating these distributions at a particular resolution
and from a given database, we assume that the distribution is everywhere Gaussian, characterized by a mean value $\mu_b$ and a
spread $\sigma_b$. We then carry out a sensitivity analysis of the simulated landfast ice cover on the spread value and the impact of
truncating the distribution. This way, the model we propose here could be applicable in a variety of configurations where the
subgrid depth information is not necessarily available, given however that the spread value is optimized.


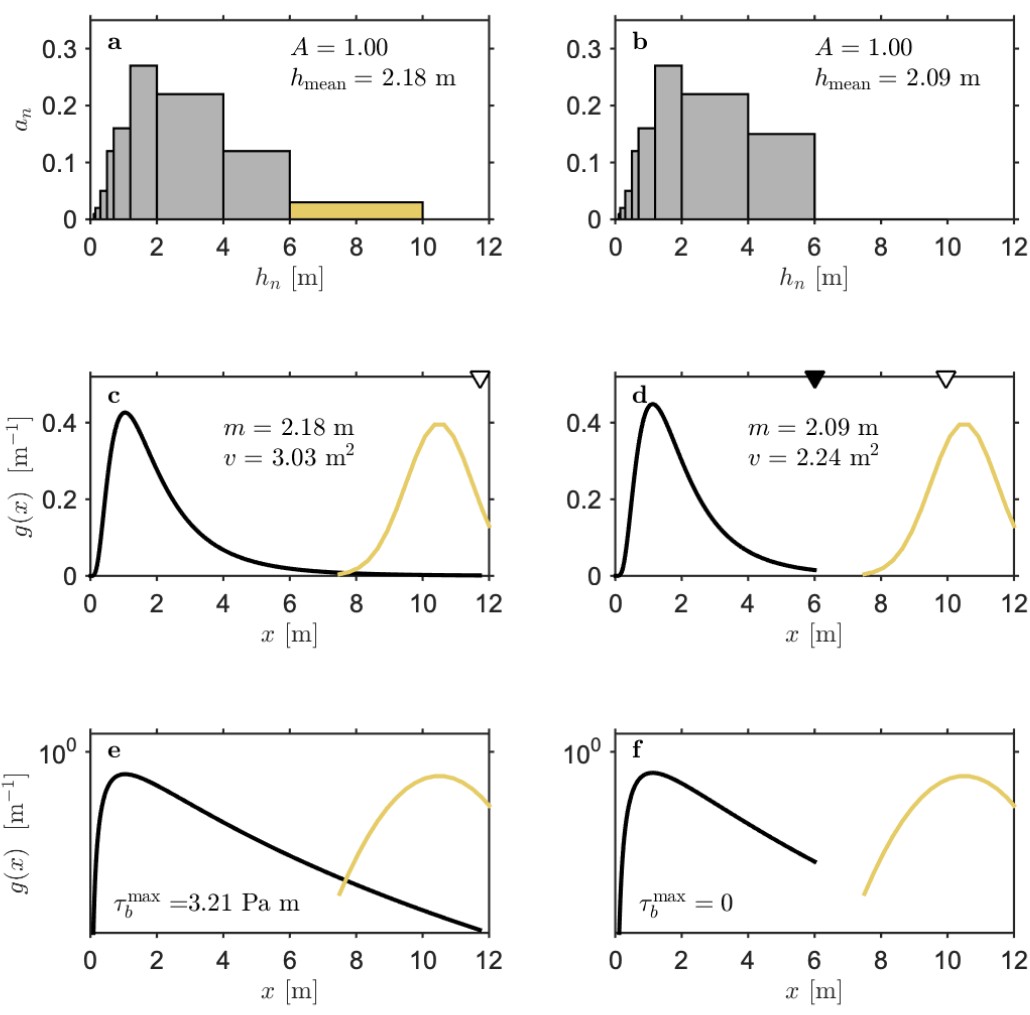

**Figure 1.** Examples showing how the maximum seabed stress due to seabed-ice friction is calculated. Total concentration $A$, mean ice thickness $h_{\mathrm{mean}}$ and partial concentrations $a_n$ of the ice thickness distribution (a,b) are used to compute the associated log-normal probability density function with same mean $m$ and variance $v$ (c,d). On the left (a,c,e), an example is shown where all categories are populated, including the last one (yellow bin) — note that for visualization purposes the last category is bounded arbitrarily at 10 m. In this case, the thickest keel draft is taken to be the 99.7[th] percentile of the PDF, which is $x_{997}$=11.73 m (open triangle). On the right (b,d,f), the partial concentration of the thickest ice category is zero. In this case, the thickest keel draft is taken to be the maximum thickness of the last category (black triangle, $x = 6$ m), even though $x_{997} = 9.96$ m (open triangle). The maximum basal stress $\tau_b^{\mathrm{max}}$ is here computed with a Gaussian bathymetric PDF with $\mu_b = 10.5$ m and $\sigma_b = 1.0$ m$^2$ and truncated at $3\,\sigma_b$ on both sides of the distribution (yellow line). The row (e,f) is the same as (c,d) but using log scale on the vertical axis



### 3.3 Parameter optimization

As L15 showed in their sensitivity study, the occurrence of landfast ice does not depend to a first order on the seabed-ice stress magnitude, which scales with the friction coefficient $\mu_s$, but rather on the keel thickness that determines when the ice is touching the seabed (i.e. the probability of contact). In the probabilistic parameterization presented here, the probability of contact in an oceanic cell, is mainly controlled by $x_p$, which is the maximum value of $g(x)$, and on the shallowest value of the bathymetry distribution $b(y)$.

#### 3.3.1 Deepest keel estimation

Let us focus first on the value of $x_p$. There are two ways we can tune this parameter. The first relies on the assumption that this value must represent the thickness of the deepest keel present over the area characterized by the ITD. Using upward-looking sonar moored in the Beaufort Sea, Melling and Riedel (1996) measured the ice draft over 941 km of drifting sea ice during winter 1991-1992. They related the draft of the keels $h_{dk}$ to the draft of surrounding level ice $h_{dl}$. The scatter plot relating these two quantities (Fig. 13 of Melling and Riedel, 1996) is a cloud of points for which the upper bound follows $h_{dk} = 16\sqrt{h_{dl}}$. Unsurprisingly, there is no unique relationship between the thickness of a keel with the thickness of the surrounding level ice. Amundrud et al. (2004) used a similar methodology using another upward-looking sonar dataset, also in the Beaufort Sea, and found a similar relationship with an upper bound that follows $h_{dk} = 20\sqrt{h_{dl}}$ (Fig. 13 of Amundrud et al., 2004).

Ideally, a sea ice model that represents ridging dynamics (keel formation) should reproduce the broad characteristics of the observed relationship. The recent work of Roberts et al. (2019) represents a comprehensive effort to improve how Earth system models reproduce ridge statistics. They do so by revisiting the energy-based method of Rothrock (1975) using variational principles, and by taking into account sea ice macroporosity through a bivariate distribution. However, in order to be generally applicable, the seabed stress formulation introduced here relies solely on the ITD. To retrieve the thickness of the deepest keels in a PDF, here using a log-normal distribution, we assume that it corresponds to a percentile that would best fit the observations. Figure 2 shows the 99.6$^{th}$, 99.7$^{th}$ and 99.9$^{th}$ percentile of the log-normal distribution, respectively called $x_{996}$, $x_{997}$ and $x_{999}$, as a function of the mean thickness for all model points on 15 April 2010 (the model will be described below). Among these three values, it appears that $x_{997}$ is the one that fit best the observations of the keel to mean thickness ratio reported by Melling and Riedel (1996) and Amundrud et al. (2004) that represent the upper bound values, with the caveat that the observations were using draft measurement, not ice thickness. This figure illustrates very well that there are indeed multiple keel depth estimates per mean ice thickness value. The choice of the percentile value has been made so that the cloud of points lies below their curves, but also by comparing the simulated landfast ice area with observations (this is discussed in section 4). In the following and otherwise noted, $x_{997}$ will be used to truncate $f_{\mathcal{LN}}(x)$.

#### 3.3.2 Seabed stress sensitivity

Figures 1c and 1d show different $f_{\mathcal{LN}}(x)$ based respectively on the ITD of Figures 1a and 1b, interacting with a truncated Gaussian PDF at $\pm 3\sigma_b$ for the bathymetry. In panel c, the thickness PDF is truncated at $x_{997}$=11.73 m, which allows for





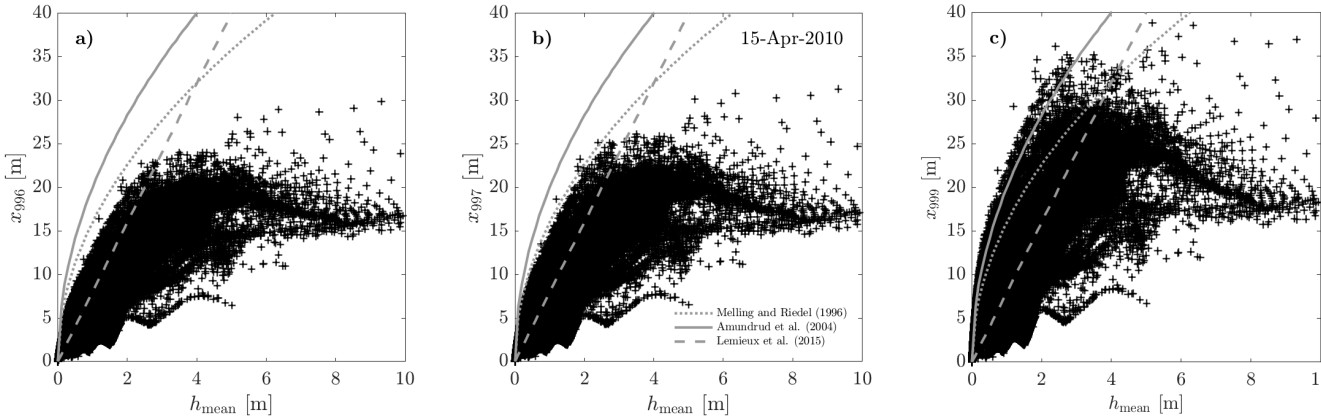

**Figure 2.** Scatterplot of a) $x_{996}$, b) $x_{997}$ and c) $x_{999}$, corresponding to the 99.6[th], 99.7[th] and 99.9[th] percentile of the log-normal distribution, as a function of the mean thickness for all model points of the domain on 15 April 2010. This is compared to the empirical upper bound determined from in situ data by (Melling and Riedel, 1996, Fig. 13) (gray line) and (Amundrud et al., 2004, Fig. 5) (dotted gray line), and with the keel depth parameterization of L15 (dash gray line).

interactions with the bathymetry, and in Panel d $x_{997}$=9.96 m but lies outside the PDF space, the maximum thickness being limited to the last bounded and populated category of the ITD. This is an illustration of the sensitivity of the seabed stress on the ITD.

The effect of the spread and truncation of the bathymetric PDF on the seabed stress is further investigated and compared to the LKD method. A full Gaussian (FG), truncated Gaussian (at $\pm 3\,\sigma_b$; TG) and truncated uniform (covering the same span as the truncated Gaussian; TU) PDFs are tested with two different values of the standard deviation. Figure 3a shows that, for $\sigma_b$=1.0 m and low values of the mean depth $\mu_b$, the new seabed stress magnitude is higher than that of LKD and that this relation reverses at higher values of $\mu_b$. Note that the figure uses a logarithmic scale and that differences are a few orders of

magnitude. As expected, TG yields a lower stress than that of the full Gaussian FG when the ice is nearly touching the seabed, but both curves converge as the depth decreases. The truncated uniform distribution (TU) yields stress values that are slightly larger as the probability of finding thicker ice is higher. Increasing the standard deviation of the depth distribution (Fig. 3b) pushes the stress values upward for bathymetry PDFs so that the crossover with LKD values happens for higher values of $\mu_b$. The stress approaches zero also later for all bathymetry PDFs in this context. The cutoff values for both truncated PDFs

are the same and nearly equal to $x_{997} + 3\,\sigma_b$, i.e. when there is no longer any contact. For $\sigma_b = 1.0$ m, the cutoff happens at a value below 14.73 m, and for $\sigma_b = 2.5$ m, the cutoff happens at a value below 19.23 m. The cutoff for LKD happens at $\mu_b = k_1 h_{\mathrm{ice}}$=16.36 m, which is visually very close to that of the truncated PDFs for $\sigma_b = 2.5$ m. As it was already noted that the actual value of the stress has less impact that the value of the deepest keels, the cutoff value is therefore an interesting evidence





that, at $\sigma_b = 2.5$ m, LKD and TG formulations should yield similar results. On the other hand, FG cutoff would be too low at
$\sigma_b = 1.0$ m and too large at $2.5$ m with a slow sloping off. In this context, we found the behaviour of TG more similar to that of
LKD and therefore more appropriate for a numerical implementation. We show further down how the landfast area is impacted
by the choice of the spread in the bathymetry distribution using TG.

### 3.4 Experimental setup

A coupled ice-ocean model is used to perform the numerical experiments. The model grid covers the Arctic and part of the
North Atlantic oceans with a nominal $0.25°$ resolution. This grid represents a subdomain of the $0.25°$ global ORCA grid, which
has a grid spacing of $\sim 12.5$ km in the central Arctic.

The sea ice model is the Community Ice Model CICE version 5.0 (Hunke et al., 2015) with some modifications: the UK
Met Office CICE-NEMO interface (Megann et al., 2014), the grounding parameterization of L15 and Lemieux et al. (2016)
and the new seabed stress parameterization described in this paper. NEMO version 3.6 (Madec, 2008) is the ocean model. It
is applied in a variable volume and nonlinear free surface configuration with 75 vertical levels, which follows closely that of
Lemieux et al. (2016). The Turbulent Kinetic Energy scheme (TKE, a simple one equation closure) is used for ocean mixing.
Both sea ice and ocean models use an advective time step of 10 minutes (600 s). The EVP method with 480 subcycles is used
to solve the sea ice momentum equation. Ten (10) thickness categories (as defined above and used first in Smith et al. (2016))
are employed for the CICE ice thickness distribution (ITD) model.

Vertical profiles of temperature, salinity and ocean currents are prescribed at the North Atlantic and North Pacific open
boundaries. These profiles come from GLORYS2 version 4 reanalysis (Garric et al., 2017) monthly-averaged fields. Atmo-
spheric forcing fields for the coupled ice-ocean simulations are the 33 km resolution reforecasts of Smith et al. (2014). This
atmospheric dataset covers the period 2001-2010. This is a relatively short period for conducting a spinup followed by an
analysis of model results. As done in Lemieux et al. (2018) and explained below, this issue is mitigated by using a special
procedure for the spinup.

Average (September-October 2001) sea ice concentration from the National Snow and Ice Data Center (NSIDC, http://
nsidc.org/data/seaice_index/) and average (October-November 2003) sea ice thickness field derived from ICESat data (https:
//nsidc.org/data/icesat) were used to initialize the sea ice model. The sea ice starts at rest. For the ocean model, the initial
temperature and salinity fields are September-October averages from WOA13_95A4 (Locarnini et al., 2013; Zweng et al.,
2013). The ocean also starts at rest; the currents and the sea surface height field are set to zero.

These fields were used to initialize the CICE-NEMO model for what we refer to as the pseudo-spinup. The pseudo-spinup
consists in a simulation running from 1 October 2001 to 30 September 2002 that is repeated three times. Following these three
years of simulation, the coupled model is restarted on 1 October 2001 and ran until 15 September 2004. These last three years
of the simulation are considered as the final spinup. Note that the LKD parameterization was used for the complete spinup
procedure. September 2004 simulated fields are used to restart simulations for optimizing parameters (see section 4) and to
perform long-term experiments for model result analyses.



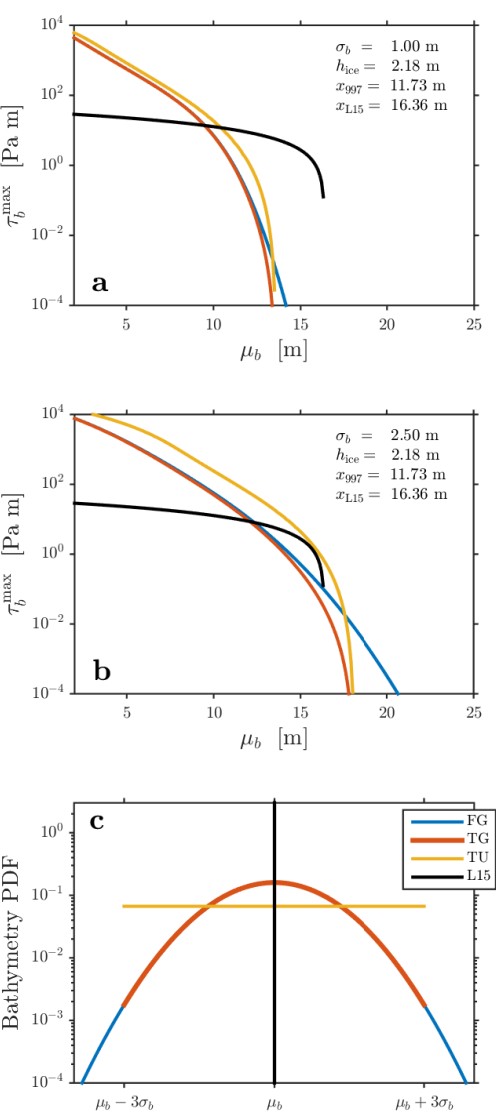

**Figure 3.** Maximum seabed stress $\tau_b^{max}$ computed with Eq. 3 using the ice thickness distribution of Fig. 1a, fitted with a lognormal distribution of mean 2.18 m and variance 3.03 m$^2$, truncated at $x_{997} = 11.73$ m, as a function of the mean water depth $\mu_b$ with spread $\sigma_b = 1.0$ m (a) and $\sigma_b = 2.5$ m (b), on a logarithmic scale. The friction coefficient is set to $\mu_s = 0.7$. The three bathymetry probability density functions (PDF) are shown in (c): a full Gaussian distribution (FG, blue line), a Gaussian distribution (TG, orange) and a uniform distribution (TU, yellow) both truncated at $\pm 3\sigma_b$. The black line in (a, b) represents the maximum basal stress of L15 with $k_1 = 7.5$ and $k_2 = 15$ N m$^{-3}$, which is independent of $\sigma_b$ since its bathymetry distribution is a Dirac delta function (black curve of bottom panel).





All the simulations for this paper use the ice strength parameterization of Hibler (1979) with $P^* = 27.5$ N m$^{-2}$. The ellipse aspect ratio is $e = 1.4$ and a small tensile strength value is added by setting $k_t = 0.05$ (Lemieux et al., 2016). Following Chikhar et al. (2019), the ice-atmosphere roughness and ice-ocean roughness are respectively set to $5.7 \times 10^{-4}$ m and $1.82 \times 10^{-2}$ m.

The other sea ice physical parameters are the default values of CICE version 5.0 (Hunke et al., 2015).

From the September 2004 simulated fields, two simulations are restarted and ran until 31 December 2010. In the first simulation, the LKD is used while in the second simulation the new probabilistic seabed-ice stress approach is used (referred to as ProbSI).

Outputs from the numerical experiments are daily mean values defined at tracer points. The daily mean ice velocity $\mathbf{u}_d = $

$u_d \hat{\mathbf{i}} + v_d \hat{\mathbf{j}}$ is used to calculate the ice speed $(u_d^2 + v_d^2)^{1/2}$ at each grid cell. As in L15 and Lemieux et al. (2016), ice is assumed to be landfast if its two-week mean speed is lower than $5 \times 10^{-4}$ m s$^{-1}$. A two-week window is used as a shorter period could cause false assessments of landfast ice when the winds are weak. The landfast ice area for a given region (the East Siberian, Laptev and Kara Seas, the three regions under consideration are displayed in Figure 4) is computed every 2 weeks by summing the area of landfast cells.

Using the McGill sea ice model, L15 found an optimal value of $k_1 = 8$ for the LKD method. A similar optimization procedure (results not shown) was repeated with our CICE-NEMO setup and led to a close value of $k_1 = 7.5$.

Finally, in ProbSI, a maximum water depth is used for limiting the sea bed-ice keel interaction to waters shallower than 50 m.

### 3.5 Landfast ice dataset

In order to determine which model formulation is the most appropriate, we compare the landfast ice area to that derived from observations, here taken as analyzed ice charts. The National Ice Center (NIC) Arctic Sea Ice Charts and Climatologies in Gridded Format dataset covers the period from 1972 through 2007 (Fetterer and Fowler, 2006, updated 2009). The analysis of landfast ice is derived from the U.S. National Ice Center weekly/bi-weekly ice charts (Detrick et al., 2001) (https://www.natice.noaa.gov/products/weekly_products.html). This product is analyzed by subject matter experts using near-

real time satellite data and various additional meteorological and oceanographic data resources, and provides observed sea ice concentration, stage of development, and partial concentration of sea ice types. Using the same approach, the landfast ice dataset is extended to cover our period of interest. We refer to our extension of the NIC dataset as NIC*ext*. Landfast ice is identified where the form of ice is coded as "08". Occasionally, landfast ice is reported with a total concentration of 10/10 without any information about partial concentration, stage of development or form of ice. The ice charts are composed of poly-

gons of various sizes and shapes. Within a polygon, it is assumed that the ice condition is homogeneous. To make use of the information, the ice charts are first rasterized on a 5-km grid. Then to compare with the model on the model grid, the nearest grid point on the 5-km grid is taken. The difference between the two landfast ice datasets (NIC and NICext) is in general very minor during the overlapping period as can be seen in Fig. 8.





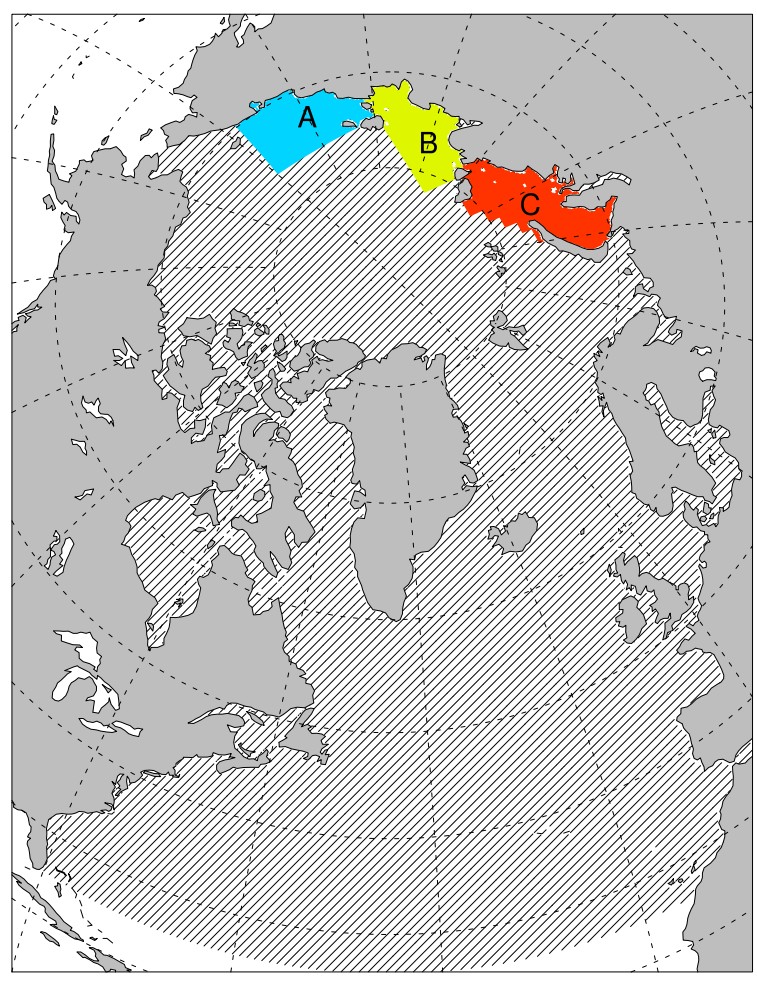

**Figure 4.** Computational domain shown in slanted black lines along with the three regions for which total landfast area is calculated ('A' for East Siberian Sea, 'B' for Laptev Sea and 'C' for Kara Sea. Taken from Fig. 2 of Lemieux et al. (2016), Crown copyrights



## 4   Results

Fig. 5 shows a pan-Arctic view of the daily averaged mean thickness $h_{mean}$, the keel depth given by $x_{997}$ in ProSI and $x_{k1} = k_1\,h_{mean}$ in LKD, as well as the difference between the two on 15 April 2010. Clearly, LKD overestimates the keel depth almost everywhere compared to $x_{997}$. The largest difference occurs north of the Canadian Arctic Archipelago (CAA) where the keel depth reaches values larger than 35 m. Even though such keel depth values have been observed on certain occasions, they still remain extremely rare and it is clearly unrealistic to have them covering such a large area (e.g. Wadhams and Davy, 1986). In

the Lincoln Sea, north of the Nares Strait that separates Canada and Greenland at 83°N, the difference between $x_{k1}$ and $x_{997}$ is larger than 20 m, owing to the fact that the ice there is thick ($h_{mean} \sim 6$ m) but relatively undeformed compared to ridging sea ice in nearby ridging lines. This stems again from the linear dependence, between $x_{k1}$ and $h_{mean}$ which overestimates the keel depth for mean thickness over 4 m (Fig. 2). The only locations where $x_{k1}$ is smaller than $x_{997}$ are in marginal ice zones and leads, where ice is generally thinner.

Note that these differences between LKD and probSI do not have any impacts except in regions where there is grounding. Over the Siberian shelves, where grounding is prevalent, $x_{k1}$ is moderately larger than the probabilistic estimation.

Figure 6a shows the weak sensitivity of running the model with the probabilistic approach and different values of the cutoff percentile at $\sigma_b = 0.5$ m in terms of the area covered with landfast ice in Region B (Laptev Sea). There are even periods where the order is not necessarily the expected one as the second lowest cutoff percentile can result in the largest area (e.g. beginning

of April 2005). The sensitivity to the variance of the depth distribution is also explored in Fig. 6b with $x_{997}$. The results closer to the observations for the maximum landfast ice cover are visually in between $\sigma_b = 1.5$ or 3.5 m. An in-between value of 2.5 is therefore chosen hereafter when not specified. This also corroborates the initial findings of 3.3.2.

In regions of recurrent landfast ice, as in the Laptev and East Siberian seas (Fig. 7), the seabed stress is compared for the same date. The ProbSI and LKD stresses are non-zero in mostly the same spots. The ProbSI stress displays more low values

(the scale is logarithmic in the Figure) which is the result of LKD having a flatter plateau followed by a sharper cutoff with increasing depth as illustrated previously in Fig. 3. Following the same reasoning but for decreasing depth, the hotpots values are larger in ProbSI. However, more importantly is that their locations — the landfast anchoring points — are the same in both formulations, which ensures a similar representation of the landfast ice.

The interannual variability in landfast area is then investigated over the different Siberian seas in Figure 8. The simulated

area in all seas is very similar using both methods and is in general very close to the one given by the proxy-observations. The two methods are especially close in Kara Sea where we expect indeed less contribution from grounding to the landfast ice formation (L15; Lemieux et al., 2016). The East Siberian Sea exhibits some bi-modal distribution of the extent of the landfast ice (the larger mode is present in 2005, 2006 and 2010 while the smaller mode is visible in almost all years) that both methods capture. Only in Kara Sea, both methods can overestimate the area and in one year in particular by almost a

factor 2. Interestingly, ProbSI exhibits breakup events in the middle of the season in Laptev Sea that would require further investigations, but otherwise is close to LKD and the proxy. ProbSI is again able to reproduce the LKD results although we note that the onset of the landfast period is always earlier in ProbSI. This means that strong modelled ridging events happen





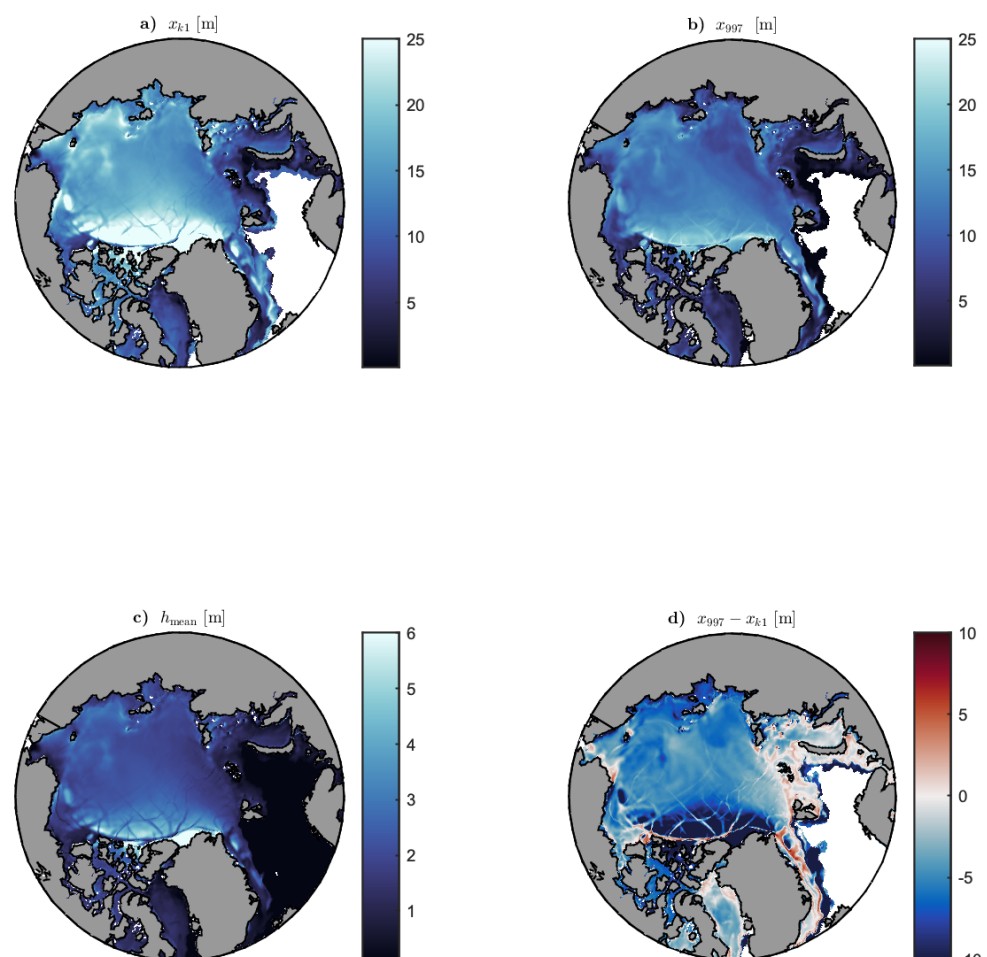

**Figure 5.** Ridge keel thickness estimated by the LKD parameterization $x_{k1} = 7.5 h_{\mathrm{mean}}$ (a) and by $x_{997}$ (b), the mean ice thickness $h_{\mathrm{mean}}$ (c) and the difference between $x_{997}$ and $x_{k1}$ (d) on 15 April 2010.



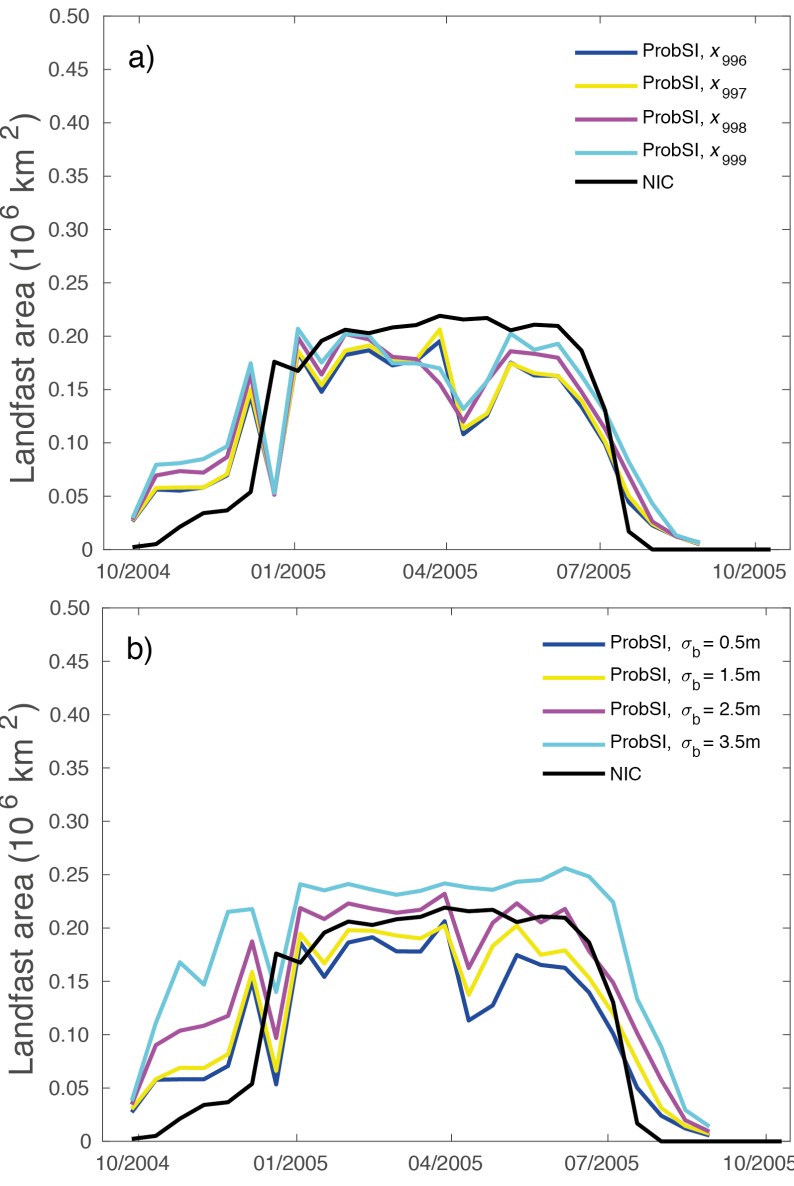

**Figure 6.** Area of landfast ice in Laptev Sea during winter 2004-2005 simulated with the ProbSI formulation and compared to NIC data. The top panel shows results with $\sigma_b = 0.5$ m and different values of $x_p$, while the bottom panel shows results with $x_{997}$ and different values of $\sigma_b$.



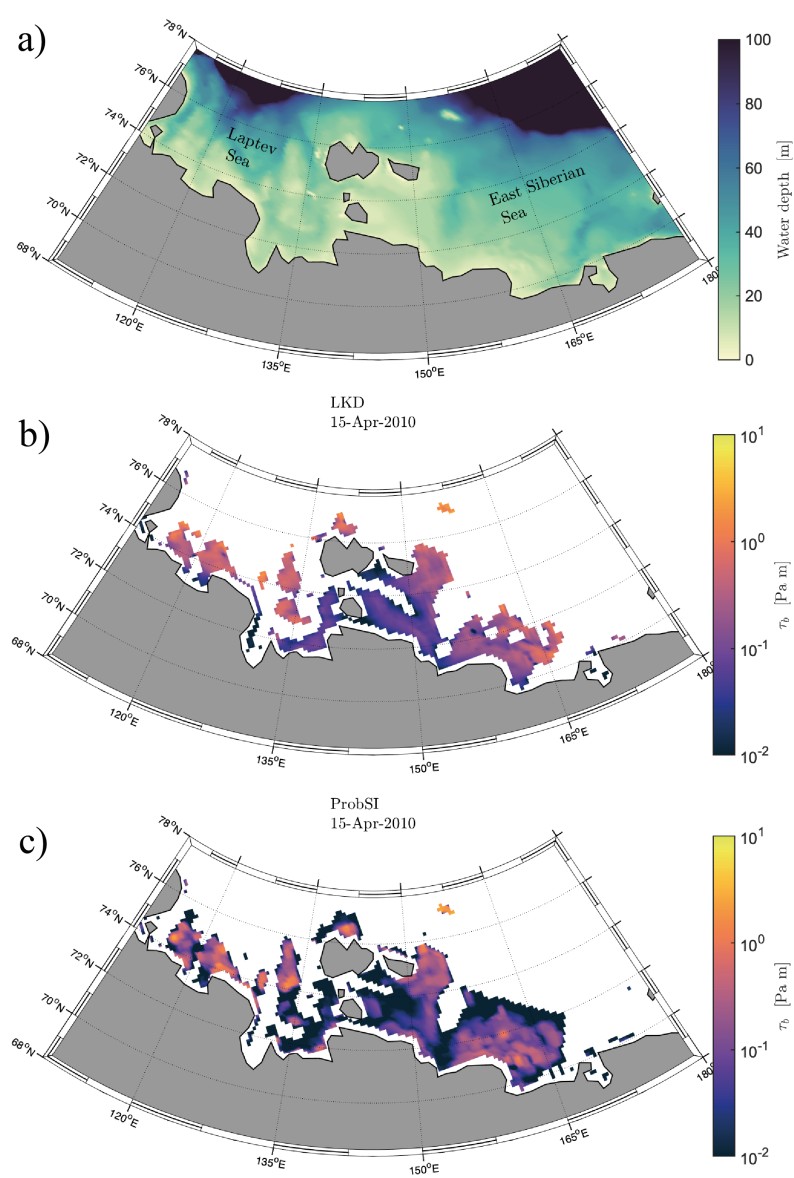

**Figure 7.** Water depth (a) and basal stress due to seabed-ice interaction in the Laptev and East Siberian Sea on 15 April 2010 for the LKD (b) and ProbSI parameterizations (c).




quite early in the sea ice season (Fig. 3). The ice chart information tends to show a slower onset, even relative to LKD. Later in the season and in the East Siberia and Laptev Seas in particular, the opposite is visible, i.e. the landfast area is larger in

LKD than in ProbSI. This indicates that, with overall thermodynamic growth, LKD responds with an associated deeper keel and larger regions of grounding whenever possible. In contrast, in ProbSI the deepest keel is more dominated by (early) local deformation, and therefore less inclined to increase during the season. The timings of the breakup of the landfast ice cover are similar for both parameterizations and agree reasonably well with the observations. However, the decrease in the simulated area of landfast ice is not as sharp as in the observations.

As introduced by Laliberté et al. (2018), we also computed the mean number of months of landfast ice per year over the six year period of the simulations (September 2004 to September 2010). Despite the stronger onset of the landfast regions in ProbSI, the overall period of landfast cover is similar to that of LKD over the Siberian shelves (Fig. 9) In particular ProbSI displays a lower number of landfast cover in the central Kara Sea, where ProbSI is more akin to breakup events as seen already in the high frequency wiggles in 8b. Nonetheless, the difference in other parts of the Arctic Ocean are very small or not

significant as in the CAA, where we expect landfast to be due mainly to the formation of ice arches (i.e. land-lock regions). We note though that both parameterizations overestimate landfast ice in some parts of the CAA because tides are not included in these simulations (Lemieux et al., 2018).

## 5  Discussion

A probabilistic approach to sea-ice grounding based on the ITD information and PDFs of the bathymetry is proposed and

compared to the simpler LKD approach. We illustrate the need for tuning one important parameter, mainly the percentile thickness at which the seabed stress calculation is truncated. Here, we selected the 97[th] percentile of the equivalent log-normal distribution, $x_{997}$, after a comparison with observations of deepest keel to mean thickness relations by Melling and Riedel (1996) and Amundrud et al. (2004), and a sensitivity analysis of the model compared with observed seasonal landfast ice cover in the Arctic.

The model simulations show a significant sensitivity to the standard deviation of the bathymetry distribution that we simply represent here as a truncated Gaussian having a value of 2.5 m over the entire domain after comparison with ice charts.

The coupled ice-ocean model is then run over the Arctic over a longer period to illustrate the difference between LKD and ProbSI methods over multiyear timescales. Results show that the onset of the landfast period happens more abruptly with the probabilistic approach and yields slightly longer landfast periods in some regions. However in general the results are very close,

i.e. within the one-month error margin associated with the definition of the landfast ice cover. The stronger onset is likely due to early deformation events captured by the model. This is however not well supported by ice chart information. An in-depth analysis is required to investigate what is behind this discrepancy. It would be understandable though that young and thin ice is easily deformed by a few stormy events, i.e. the grounding must be quick to happen once new ice is present and that the model does not necessarily overestimate those events. By refitting the ITD to a log-normal distribution using the mean and variance,

it is expected that the ProbSI method may be less reliable when the ice is thin. For example, forming new thin ice might in





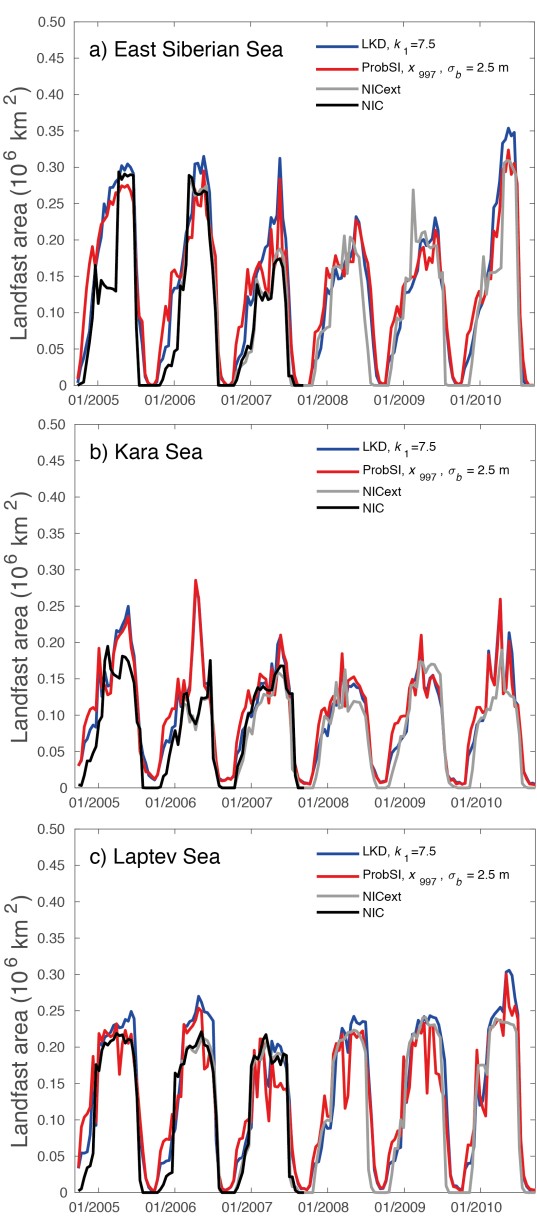

**Figure 8.** Area of landfast ice in the (a) East Siberian Sea, (b) the Kara Sea, and (c) the Laptev Sea as a function of time. The blue curves are for LKD while the red ones are for $x_{k\mathrm{max}} = x_{997}$ and $\sigma_b = 2.5$ m. In the three figures, the bold black curve is the area calculated from the NIC data while the gray curve is extended database produced with the same method.

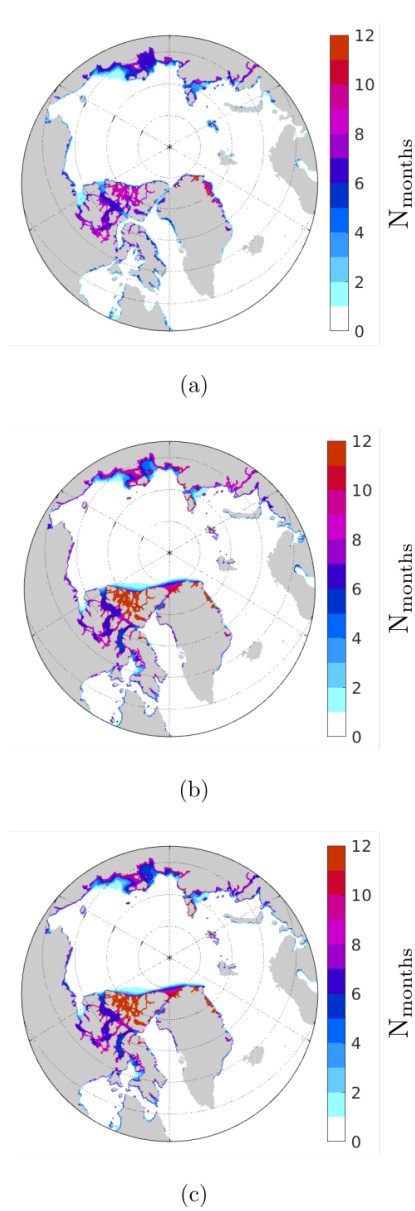

**Figure 9.** A 6-year average of the number of months per year during which the ice is landfast obtained from the NIC charts and extensions (a), with the LKD parameterization (b) and the ProbSI model with $\sigma_b = 2.5$ m (c)

some cases lower the variance thereby lowering the keel depth and reducing the friction with the seabed. This is maybe what explains break-up events in ProbSI that do not happen in LKD in the Laptev Sea (see in Fig.8).

Nonetheless, because this method uses a somewhat more realistic representation of maximum keel depths, discrepancies between results and observations are prone to highlight problems or caveats in other aspects of the model, such as sea ice

dynamics, ridging schemes, ITD resolution, or insufficient or bad bathymetry data in some areas. In particular, we stress the importance of retaining sub-grid-scale bathymetry information in order to better constrain $\mu_b$ and $\sigma_b$.

Obtaining $x_{997}$ as an estimation of the deepest keel depth was done by looking at relatively thick and highly deformed ice. The discrepancies noted between LKD and ProbSI in marginal ice zones (c.f. Fig 5d.) were not investigated, as it was not the focus of the paper, and should be interpreted with care. Sea ice and dynamics and thickness redistribution is potentially

influenced by processes associated with surface ocean waves propagating into the ice cover such as fragmentation and rheology (Dumont et al., 2011; Boutin et al., 2020) that are not included yet in CICE. Nonetheless, the fact that $x_{997}$ is significantly different than $x_{k1}$ in Fram Strait, a region where highly deformed ice cohabits with new ice formed in leads and cracks, is consistent with the fact that the maximum keel depth is a nonlinear function of the mean ice thickness, which is duly taken into account by ProbSI.

The ProbSI method for computing the seabed-ice keel interaction is implemented in CICE version 6.0 and is available at https://github.com/CICE-Consortium/CICE.

*Code and data availability.* The National Ice Center landfast ice data are available at http://nsidc.org/data/G02172. The National Ice Center weekly/bi-weekly ice charts are available at https://www.natice.noaa.gov/products/weekly_products.html. The CICE-NEMO code and the atmospheric forcing used for the simulations are available upon request.

*Author contributions.* FD coordinated the manuscript and wrote sections; DD initiated the research and wrote sections; JFL ran the computer simulations and coordinated the CICE code changes; EDL wrote the initial code in matlab and ran the initial tests as part of his master thesis and took great care of many figures; AC provided the landfast ice data used for comparison

*Competing interests.* no competing interests are present

*Acknowledgements.* DD would like to thank Anne-Claire Bihan-Poudec, Simon Senneville and Zakaria Belemaalem who helped develop
the idea and first applied it in the Gulf of St. Lawrence. This work was financially supported by the NSERC Discovery Grant to DD *Physics of Seasonal Sea Ice* (RGPIN-2019-06563).



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
