# Peer review of "A probabilistic seabed-ice keel interaction model"

_The Cryosphere, 2021_

## Author Comment (AC1)

Reply to Andy Mahoney comments on

**A probabilistic seabed-ice keel interaction model** by
Frédéric Dupont, Dany Dumont, Jean-François Lemieux, Elie Dumas-Lefebvre and
Alain Caya

Comments are reproduced in italic text and followed by our response in normal text
and in blue.

**Major comments**

*1. Mismatch between mean ice thickness and level ice thickness*

*Near the beginning of Section 3.1.1, the authors note that both Melling and Riedel (1996) and Amundrud et al (2004) derived relationships between keel depths (hdk) and the draft of surrounding level ice (hdl). However, later in the same paragraph the text describes these relationships as being between keel depth and mean ice thickness (hmean). I thought that this may have been a simple typo, but the x-axes in Figure 2 are labelled as hmean and the text in the discussion on lines 347-348 again refers to mean ice thickness. I am therefore concerned that the authors may be incorrectly applying the findings of Melling & Riedel and Amundrud et al by applying their relationships to mean ice thickness instead of level ice thickness. Since mean thickness will almost always be greater than the level ice thickness in a grid cell, this will have the effect of moving the curves shown in Figure 2 downward, suggesting that x997 may not be the best fit as claimed.*

The issue that is raised here is quite relevant and was raised during the production of this manuscript. We agree that there is an inconsistency between the mean ice thickness and the *thickness of level ice surrounding a ridge*. We used the mean ice thickness first because there was no agreed upon definition of level ice thickness in the model output, and second, because the choice of the percentile value was not solely determined by the fit with the empirical relationships. Instead, the choice of the percentile evalue mostly relies on the comparison between the observed and simulated landfast ice cover area (Figures 6 and 8).

Despite this, we agree that our current Figure 2 and the associated text are misleading. To correct this, we propose a new figure below that shows a more adequate, yet not perfect, metric for level ice using only the available 10-category ITD. Level ice is defined by Melling et al. (1996) as the thickness of the ice flanking the associated ridge. We can't isolate individual ridges from the model ITD. We can assume however that the last category ($h > 4$ m) represents only ridged ice, and that ice going into the last category through ridging comes from ice from categories

below. We can thus assume that the average thickness of the ice thinner than 4 m is somewhat representative of the ice surrounding the thickest ridges. We thus define $h_{level}$ as the mean ice thickness of the ice thinner than 4 m, and $h_{mean}$ the overall mean ice thickness. The relation between those two quantities will bear a similar meaning as in Melling et al. (1996) only if we consider fully compact ice, hence with $A > 0.9999$. The new Figure 2 below shows a) the $h_{level}$-to-$h_{mean}$, a) the $x_{997}$-to-$h_{mean}$, and c) the $x_{997}$-to-$h_{level}$ relationships using the proxy described above. Blue dots are all the points while black dots show only those with $A > 0.9999$. Panel a shows that level ice thickness as a function of mean ice thickness is scattered, unsurprisingly always below the 1:1 curve and saturates when $h_{mean}$ increases for values above 4 m. Panel c shows that $x_{997}$ is still a good predictor of the largest keel depth when compared to empirical relations of Amunrud et al. (2004) and Melling et al. (1996). The text has been modified to accommodate this change. Figure 2 is not presented anymore as a key tuning procedure for $x_p$ but rather a confirmation that the probabilistic approach based on the log-normal ITD compared reasonably well with available observations.

[Figure]

Figure 2. a) Relationships between level ice thickness, defined as the mean thickness of ice thinner than 4 m. b) The largest keel thickness $x_{997}$ as a function of the mean ice thickness $h_{mean}$. The gray dash line represents the L15 parameterization. c) x997 as a function of hlevel, with the solid and dash lines showing empirical relationship of Melling et al. (1996) and Amunrud et al. (2004), respectively. Blue dots represent all grid points for 15 March 2010 while black dots show only those with $A > 0.9999$.

**2. Lack of clarity in key figures**

*Figures 1 and 3 are important figures, but both could use work to improve their usefulness to the reader. I found it necessary to read both the captions and the main body of text multiple times before I understood what either figure was supposed to be showing. Although I appreciate the avoidance of what Tufte (2001) describes as "chart junk", I believe the information content of each figure as a whole would be*

*greatly improved with better labelling. Specifically, I would recommend adding a legend to Figure 1 to explain the meaning of each curve and symbol without having to read a full-paragraph caption. This would also allow the caption to be shortened significantly.*

*I also recommend using textual axis labels so that the reader doesn't have to refer back the main text to remember what each symbol or abbreviation means. For example, the yaxis in Figure 3c is labeled "Bathymetric PDF", rather than b(y), and I recommend adopting this practice for all axes and legends. Also, for accuracy, the y-axes of Fig 1c-f should reference both ice thickness and bathymetry.*

*Lastly, I recommend using a different color to highlight the final ice thickness category in Figure 1a, since the choice of yellow suggests some relationship to the yellow curves in panels c-f.*

The new Figure 1 below implements these recommendations.

[Figure]

The new Figure 3 has also been modified slightly.

[Figure]

**3. Incomplete discussion of difference in landfast ice development in Laptev Sea**

*The authors draw attention to the earlier and more rapid development of simulated landfast ice in the Laptev Sea, as compared with observations of landfast ice in ice charts from the U.S. National Ice Center. They state that an "in-depth analysis is required to investigate what is behind this discrepancy" (lines 356-357), but suggest that it may be related to overestimated of keel depths resulting from deformation of thin ice. Although I do not want to suggest any new in-depth analyses, I would recommend additional discussion referring to the work Selyuzhenok et al (2015; 2017), which describes the formation of landfast ice in the Laptev Sea in some useful detail.*

*Specifically, Selyuzhenok et al (2015) identify a period of "initial formation" November and December, during which time landfast ice slowly approaches approximately 20% of its annual maximum extent. This is followed by brief period of rapid expansion when most of the remaining expansion takes place. These periods are robust features of the annual cycle of landfast ice in the Laptev Sea and are captured in the NIC-derived landfast ice extent shown in Figure 6. In their 2017 paper, Selyuzhenok et al go on to show that during the initial formation period, grounded features can form offshore while being entirely surrounded by ice that is still mobile. The drift speed of the mobile ice gradually decreases until the ice becomes stationary, at which point there is a rapid growth in landfast ice extent. Selyuzhenok et al (2015) attribute the onset of rapid growth to the achievement of a critical thickness or strength within the formerly mobile ice. Hence, the ProbSI model may not be overestimating keel depths, but instead overestimating the shear strength of the surrounding ice.*

Thanks for the two references and the suggestion. The first reference appears the most relevant to our paper. The following sentence was added in the discussion section to highlight the suggestion:
"Selyuzhenok et al. (2015) suggest another explanation, namely that a stronger than observed resistance of the model to deformations can also contribute to the landfast ice formation. Another suggestion in \citet{Selyuzhenok2017} is that the ice can still be mobile despite the presence of ridged features anchored to the bottom."

**Minor comments**

*Line 27: Replace "Most" with "More"*
Done.

*Line 112: I assume that the symbol σ in equation 5 refers to the internal stress within, but since not all readers will be familiar with the sea ice momentum equation, it*

*should be explicitly defined. Also, it appears that σ is used later in a different context (see comment for line 158), so further clarification maybe needed.*
Done, referred as \sigma_i instead.

*Lines 118-119: I believe the cross reference to section 3 should be a reference to 3-pointsomething*
Done. It's 3.2.

*Line 158: σ is apparently being used here to a different property than in equation 5 above (see comment for line 112). A different symbol should therefore be used either here or above. Also, I recommend providing a physical explanation of both σ and μ as expressed here.*
Done.

*Line 176-177: How are σb and μb related to σ and μ as defined in equations 7 and 8? Here, σb and μb are referred to as "mean value" and "spread". Is this how σ and μ should be interpretted?*
Yes, sigma_b and mu_b are the mean value and spread (or standard deviation) of the Gaussian distribution. We have changed "spread" for "standard deviation" in the text when applicable to the bathymetry distribution.

*Line 205: I recommend replacing "This figure" with "Figure 2"*
Done.

*Line 221: Where the text reads "the depth", I assume the authors are referring to water depth. However, since the text regularly refers to both water depth and keel depth, I recommend taking care to specific each time the term depth is used*
Done. This was checked for the whole manuscript and equivocal instances of "depth" were replaced by "water depth".

*Lines 221-222: I'm confused here. Please explain why the probability of finding thicker ice is greater with the TU bathymetric distribution*
The expression was indeed misleading. It should have read `probability of shallow bathymetry intersecting ice`. Modified in the text.

*Line 224: The use of "later" here is imprecise as it suggests a time-dependent process. I believe a phrase like "at a greater mean water depth" would be more appropriate.*
Indeed, done.

*Line 227: I find the phrase "visually very close" to be ambiguous. I recommend finding a more accurate and specific phrase.*

Replaced by 'qualitatively'.

*Line 228: I believe "less impact that" should read "less impact than". Also, I assume the authors are referring the impact on basal shear stress, in which case I think it would help to add "on basal shear stress" after "impact".*
Done.

*Figure 3: See major comment 2 above regarding the replacement of abbreviations in the legend with full text. Also, I believe the word "truncated" is missing on the 4th line of the caption before the second usage of "Gaussian".*
Please see above for response to comment 2 and abbreviations. "truncated" added at the corresponding place.

*Line 335: I am not convinced that the number of months of landfast ice per year is a meaningful metric when the timing of formation and breakup is not well reproduced. This approach would suggest the model is somehow more accurate if it simulates earlier dates of both formation and break up. Those are two separate errors that this metric will mask.*
The reviewer is correct on principle although it is doubtful that the model is too early for both formation and melt as models tend to have a more systematic bias of being either too warm (ice formed later and too thin; therefore melting too fast) or too cold (the opposite). Thus, that metric is not necessarily easier on the models. We are also trying to stay consistent with the metric used in other papers (e.g., Laliberté et al. 2018).

*Line 336: I do not know what "stronger" means in the context of landfast ice onset. I recommend using plainer, clearer language. In this case, I think "earlier", or "more rapid" (or perhaps both) would be more appropriate.*
We suggest 'earlier' then.

*Line 338: Please clarify what "lower number of landfast ice cover means". I think there maybe a typo here, but I can't uniquely identify a solution*
Sorry, it should have read 'months'.

*Line 338: I don't think "akin" is the right word here. Perhaps "prone" or "susceptible" would be more appropriate.*
I agree, 'prone' sounds better.

*Line 339: While appreciate the graphically descriptive nature of the term "high frequency wiggles", I am sure the authors could find a phrase that more accurately describes the variability to which they are drawing attention.*
We used high frequency fluctuations.

**References cited in this review that are not cited in the submitted manuscript**

Selyuzhenok, V., T. Krumpen, A. Mahoney, M. Janout, and R. Gerdes (2015), Seasonal and interannual variability of fast ice extent in the southeastern Laptev Sea between 1999 and 2013, Journal of Geophysical Research: Oceans, 120(12), 7791-7806, 10.1002/2015JC011135.

Selyuzhenok, V., A. Mahoney, T. Krumpen, G. Castellani, and R. Gerdes (2017), Mechanisms of fast-ice development in the south-eastern Laptev Sea: a case study for winter of 2007/08 and 2009/10, Polar Research, 36(1), 1411140, 10.1080/17518369.2017.1411140.

Tufte, E. (2001), The visual display of quantitative information, edited, Cheshire: Graphic Press.–2001.–213 p.

---

## Author Comment (AC4)

Reply to RC2 comments on

**A probabilistic seabed-ice keel interaction model** by
Frédéric Dupont, Dany Dumont, Jean-François Lemieux, Elie Dumas-Lefebvre and
Alain Caya

Comments are reproduced in italic text and followed by our response in normal text
and in blue.

*The manuscript describes extended version of grounding scheme by Lemieux et al.,
2015. Authors provide theoretical description of the method, apply it for short term
sea ice-ocean simulations and describe the results.*

*The paper is very well written, and enjoyable to read. Figures are also of a good
quality. I have only several very small comments, and in my view, paper can be
accepted after minor revision.*

***Minor comments:***

*Line 30. It would be nice to see a paragraph about other attempts to add fast ice in
the Arctic Ocean simulations, like Lieser et al., 2004, Itkin et al., 2015 and Olason,
2016.*

Done, the three references are discussed in the introduction:

"Various attempts at representing landfast ice in Arctic simulations have been
conducted in the past with a mix of these processes, starting from the crude zero
velocity condition using an ice thickness-to-depth ratio of \citet{Lieser2004}, an
increased maximum viscosity in \citet{Olason2016}, an artificially large tensile
strength in \citet{Itkin2015} or the seabed stress parameterization of \citet[][hereafter
referred to as L15]{Lemieux2015}."

*Lines 38-39: Please comment on computational efficiency as well.*

The additional computational costs associated with the LKD method is very small
compared to the rest of the sea ice model computations (thermodynamics, dynamic
solver, transport, ridging, etc...) as the main calculation is the maximum seabed
stress (computed outside of the EVP subcycle) and the remaining operations during
the subcycling are minor. We feel though that this explanation does not need to
appear in the manuscript.

*Line 57. You probably mean then --> than.*

Thanks, corrected.

*Line 82. While it became obvious from the rest of the paper why you represent
bathymetry as random variable, a simple additional sentence giving the motivation*

*for it would be useful for ocean modelers like me, who often just take bathymetry as something that is well defined.*

Done. Adcroft (2013) shows actually an interesting use of the subgrid scale bathymetry information to restrain the bottom flow. Reference added in the manuscript.

Adcroft, A. (2013). Representation of topography by porous barriers and objective interpolation of topographic data. Ocean Modelling, 67, 13-27.

*Line 118. …here (see Section 3) --> in this section*

Following the first reviewer, we went for "section 3.2", for more precision.

*Line 119. "The following SUBsections".*

Thanks, done

*Line 163. You mean Subsection 3.3.1 here, I guess.*

Yes, thanks for the correction

*Line 242. Why so many EVP cycles? The standard value for CICE is around 120, if I am not mistaken?*

Ah, thanks for bringing this topic up. There is in fact mounting evidence that the 120 is way too low (Lemieux et al., 2012; Kimmritz et al. 2015, Xu et al., 2021) if you want to approach the viscous-plastic solution, which is the underlying assumption behind the EVP solver, although the number seems to be also a function of resolution. Note that the standard value in CICE6 is now 240 (which we believe is still too small!)

Lemieux, J. F., Knoll, D. A., Tremblay, B., Holland, D. M., & Losch, M. (2012). A comparison of the Jacobian-free Newton–Krylov method and the EVP model for solving the sea ice momentum equation with a viscous-plastic formulation: A serial algorithm study. Journal of Computational Physics, 231(17), 5926-5944.

Kimmritz, M., Danilov, S., & Losch, M. (2015). On the convergence of the modified elastic–viscous–plastic method for solving the sea ice momentum equation. Journal of Computational Physics, 296, 90-100.

Xu, S., Ma, J., Zhou, L., Zhang, Y., Liu, J., & Wang, B. (2021). Comparison of sea ice kinematics at different resolutions modeled with a grid hierarchy in the Community Earth System Model (version 1.2. 1). Geoscientific Model Development, 14(1), 603-628.

*Line 247. Please comment on what is the advantage of this forcing, which seem to be popular in regional ocean modelling, but is quite exotic for global modelling.*

This forcing has indeed become the reference for all the experiments done in our group. One interesting feature is the enhanced resolution in space and time while maintaining a fair degree of precision (see the provided reference, Smith et al., 2014, for more precisions). We feel though that this explanation does not need to appear in the manuscript.

*It would be good if you mention computational efficiency of the scheme in Section 3.4. Just if it decreases the model speed to a noticeable amount.*

Indeed, both methods do not decrease speed to a noticeable amount. As explained above, for both methods, the computation of the maximum sea bed stress is done outside of the EVP subcycling, while the rest of the operations inside the subcycling are very minor. We feel though that this explanation does not need to appear in the manuscript.

*Line 325. "… a factor OF two".*

Thanks corrected.

**Discussion**

*The resolution in the model setup is around 12.5 km in the Arctic. Please comment on how well, you think, this grounding scheme will be working in higher resolution setups (e.g. ORCA12 and higher).*

Indeed 12 km is not considered anymore a high resolution, but it is sufficient to resolve well enough the Siberian Shelves and to a lesser degree the Alaskan and Canadian Shelves. The proposed probabilistic scheme has the advantage that the subgrid topography is taken into account. Therefore, the actual resolved topography has less importance than its subgrid distribution. With higher resolution, we expect that the mean depth will be better represented but, given a realistic subgrid distribution and that ridges are not represented explicitly (i.e., the continuum approximation of sea ice), the sea ice dynamics and landfast ice position should be relatively invariant to the resolution. We feel though that this explanation does not need to appear in the manuscript.

*Please add to the discussion comparison to other studies, that try to simulate fast ice.*

Done in the introduction (see above response).

**References**

*Lieser, J. (2004), A numerical model for short-term sea ice forecasting in the Arctic (Ein numerisches Modell zur Meereisvorhersage in der Arktis), Rep. Polar Mar. Res. (Berichte zur Polar Meeresforschung), vol. 485, 93 pp.*

*Einar Olason, A dynamical model of Kara Sea landâ fast ice, Journal of Geophysical Research: Oceans, 10.1002/2016JC011638, 121, 5, (3141-3158), (2016).*

*Itkin, P., Losch, M., and Gerdes, R. (2015), Landfast ice affects the stability of the Arctic halocline: Evidence from a numerical model, J. Geophys. Res. Oceans, 120, 2622– 2635, doi:10.1002/2014JC010353.*
*Citation: https://doi.org/10.5194/tc-2021-273-RC2*